# SEMANTIC STRUCTURE IN LARGE LANGUAGE MODEL EMBEDDINGS

## ABSTRACT

Psychological research consistently finds that human ratings of words across diverse semantic scales can be reduced to a low-dimensional form with relatively little information loss. We find that the semantic associations encoded in the embedding matrices of large language models (LLMs) exhibit a similar structure. We show that the projections of words on semantic directions defined by antonym pairs (e.g. kind - cruel) correlate highly with human ratings, and further find that these projections effectively reduce to a 3-dimensional subspace within LLM embeddings, closely resembling the patterns derived from human survey responses. Moreover, we find that shifting tokens along one semantic direction causes off-target effects on geometrically aligned features proportional to their cosine similarity. These findings suggest that semantic features are entangled within LLMs similarly to how they are interconnected in human language, and a great deal of semantic information, despite its apparent complexity, is surprisingly low-dimensional. Furthermore, accounting for this semantic structure may prove essential for avoiding unintended consequences when steering features.

## 1 INTRODUCTION

Large language models (LLMs) display an extraordinary ability to mimic human linguistic behavior, but the extent to which the models' internal representations resemble human cognitive models remains unclear Ku et al. (2025). On one hand, it is plausible that LLMs, being trained on extensive records of thought, behavior, and interaction, would build internal representations closely mirroring those of the humans who produced these training data. Yet on the other hand, LLMs utilize a different architecture than the human brain, their training data is qualitatively different from the stimuli humans receive during development, and their task of "next token prediction" may fundamentally differ from the objectives of human learning. Improving our understanding of the representation of meaning within LLMs not only has scientific value, but may also be valuable for practical applications relating to model safety, auditing, and control Shah et al. (2025); Zou et al. (2023).

In this paper, we take a longstanding finding from social psychology and investigate its relevance to LLM internal representations. Specifically, a long literature finds that human ratings across diverse semantic scales tend to follow a strong and systematic correlational structure; for example, things that are considered "soft" tend to also be labeled as "kind," things that are "strong" tend to be "big." By consequence, ratings on a wide set of semantic attributes can be effectively reduced to a three-dimensional solution with relatively little information loss Osgood et al. (1957); Heise (2010). Research in the Semantic Differential tradition identifies these three latent dimensions as Evaluation (good vs. bad), Potency (strong vs. weak), and Activity (moving vs. stationary), and other lines of research find similar latent factors, like Warmth and Competency Fiske et al. (2018) or as Valence, Arousal and Dominance Barrett & Russell (1999).

Using techniques developed with word embedding models and extended successfully to LLMs Garg et al. (2018); Park et al. (2023), we extract feature directions from LLM embedding matrices corresponding to 28 key semantic axes (e.g. kind-cruel, foolish-wise). We project vectors for individual words (tokens) onto these feature vectors and show that these projections correlate highly with human ratings of those words on the respective semantic scales. Having confirmed the correspondence between token projections and semantic associations, we apply principal components analysis (PCA) to the projections and find that a 3-dimensional solution preserves between 40 and 55% of

the variance across 28 original features, and that the loadings on these principal components imply a structure similar to the Evaluation, Potency, and Activity dimensions identified in prior research with human subjects.

Having identified this semantic structure, we consider the implications of feature alignment for model behavior and steering. Specifically, we hypothesize that intervening on one feature is likely to have predictable off-target effects on other features proportional to their cosine similarity. For example, if *soft-hard* is closely aligned with *kind-cruel*, we expect interventions on *soft-hard* to have a stronger off-target effect on this direction than it would on an orthogonal feature, like *foolish-wise*. To test this hypothesis, we prompt LLMs to report semantic associations for a set of words, just as respondents would do in a psychological questionnaire. After collecting baseline data on LLM semantic associations for the set of words, we intervene on the model's token embeddings, steering the respective word vectors in the direction of one semantic feature, then measure the effect on reported associations for all other semantic features. Our results support the hypothesis that the magnitude of off-target effects is proportional to the cosine similarity between the target and off-target feature vectors.

Our study makes several contributions:

1. We find that semantic features in LLM embeddings exhibit a strong and meaningful correlational structure. The number of "almost orthogonal" vectors that can be represented in an $n$-dimensional space is exponential with respect to $n$ Vershynin (2018). This implies that LLM embeddings with $n > 1000$ are capable of representing a tremendous number of approximately orthogonal features. Despite this theoretical possibility, we find substantial correlation between many features critical for expressing cultural association and meaning.

2. This feature structure closely mirrors patterns commonly found in human semantic ratings, suggesting that the correlation of features is not an artifact of the embedding representation, but a meaningful characteristic of human language and understanding.

3. Finally, we demonstrate that this structure has important implications for efforts to isolate and steer individual features. Previous studies have identified such "off-target effects" but have generally treated them as random Durmus et al. (2024); Park et al. (2023). We show that the correlation structure of features is predictive of these off-target effects, an insight that could help anticipate and mitigate unintended consequences of feature steering.

## 2 BACKGROUND

### 2.1 LARGE LANGUAGE MODEL EMBEDDINGS

Autoregressive decoder-only LLMs consist of a stack of transformer blocks bookended by an embedding matrix at the beginning and an unembedding matrix at the end Vaswani et al. (2017); Radford et al. (2019); Grattafiori et al. (2024); Kamath et al. (2025); Yang et al. (2024). The embedding matrix at the beginning of the model plays the role of mapping discrete tokens to vectors in a distributed representation. These vectors are then passed through the transformer stack, after which they are mapped back to tokens by the unembedding matrix.

While most existing LLM interpretability research has focused on activation patterns in the models' transformer blocks Bricken et al. (2023); Lindsey et al. (2025); Tigges et al. (2023); Arditi et al. (2024), embedding and unembedding matrices also warrant attention. First, because vectors in embedding matrices are directly associated with individual tokens, directions in these spaces are more easily linked to interpretable concepts than vectors in the model's transformer blocks, which are farther removed from specific tokens. Second, embedding matrices are comprised of *weights*, not activations. This means that semantic relations identified in these matrices are not conditional on specific input texts, but instead influence all inferences.

### 2.2 FEATURE ENTANGLEMENT AND SUPERPOSITION

Deep neural networks encode a multitude of "features," or concept representations, but these features rarely correspond to individual neurons. Instead, features tend to be represented as direction or region in the model's latent space specified by a combination of activations across many neurons

which are used in other combinations to represent different features Bricken et al. (2023); Park et al. (2024) This phenomenon, recently dubbed "superposition" is hypothesized to emerge because $n$-dimensional spaces can represent $\gg n$ nearly orthogonal vectors when $n$ is sufficiently large Elhage et al. (2022). Therefore, the benefit of representing more features typically outweighs the cost incurred by the minimal interference between slightly non-orthogonal feature vectors. Thus, a neuron that is associated with "car" may also be highly associated with "cat", but the specific feature that is ultimately activated depends on how this neuron activates in combination with other neurons Olah et al. (2020).

Superposition is often described as an emergent method for packing more features into a model than there are dimensions, with the consequence that random sets of features may end up in superposition Elhage et al. (2022). However, a long literature on distributed representations suggests that features can be *meaningfully* non-orthogonal, and that substantial alignment is likely to occur between features based on their semantic relations Mikolov et al. (2013b;a); Pennington et al. (2014). Indeed, in their foundational work on distributed representations, Hinton, McClelland and Rumelhart Hinton et al. (1986) argued that using orthogonal representations for meaningfully related features would eliminate "one of the most interesting properties of distributed representations: They automatically give rise to generalizations" (p.82) Hinton et al. (1986).

From this perspective, distributed representations perform a kind of dimension reduction, representing $k$ observed features as linear combinations of a smaller number $n$ latent features with minimal distortion or loss of information Levy & Goldberg (2014). This interpretation has been particularly fruitful for analyses of shallow neural networks like word2vec, where the representations' basis directions do not correspond to interpretable features but words with similar meanings cluster together and directions in the space correspond to continuous features like *good-bad*, *masculine-feminine*, or *rich-poor* Garg et al. (2018); Mikolov et al. (2013b;a); Pennington et al. (2014); Kozlowski et al. (2019); Caliskan et al. (2017); Bolukbasi et al. (2016). Indeed, the reason why word2vec and related algorithms garnered so much attention because meaningful relations between words are systematically encoded as proximity in the embedding space. Yet, despite widespread recognition that angles between features preserve semantic relations in classical word embeddings, recent analyses of LLM features has paid little attention to these geometric relations, except where they form polytope structures that approximate orthogonality Park et al. (2024); Elhage et al. (2022).

If semantic features are meaningfully non-orthogonal in LLMs, this could complicate efforts of "feature steering," in which the user amplifies or mutes a feature by isolating and modifying it in the model internals. Recent efforts to control LLM outputs and reduce unwanted behaviors through feature steering do indeed report unintended "off-target" effects on other features when the target feature is modified Durmus et al. (2024); Park et al. (2023). However, this literature has viewed off-target effects as primarily a consequence of random superposition of features and has made little effort to identify systematic patterning to these effects. Prior studies attempt to minimize off-target effects by reducing the alignment between features. Park et al. Park et al. (2023), for example, apply a whitening operation to unembedding vectors so that intervention on one feature induces minimal off-target effects on others. While minimizing feature correlation does reduce off-target effects, we hypothesize that forcing orthogonality may distort the representation of the underlying concepts if features are meaningfully non-orthogonal and their alignment reflects real semantic relations. That is to say, if the alignment of features in the model's latent space encodes semantic information, then non-orthogonality may be a "feature" and not a "bug" in distributed representations.

## 2.3 SUBSPACE OF CULTURAL SENTIMENTS

The entanglement of concepts has been a longstanding subject of interest in psychology. The 1950s saw the development of the Semantic Differential approach for measuring cognitive associations, in which respondents rate a list of words on a set of semantic scales such as warm-cold, soft-hard, and ugly-beautiful Osgood et al. (1957). Scholars using this methodology arrived at two key findings. First, they found that human ratings of hundreds of items across dozens of scales could consistently be reduced to a 3-dimensional subspace with surprisingly little loss of information. Second, they found that the same latent factors, roughly corresponding to Evaluation (good vs. bad), Potency (strong vs. weak), and Activity (active vs. passive), emerged from analyses of a wide variety of countries and cultural groups, suggesting that the three-factor structure may be a human semantic universal Osgood (1964); Heise (2010). These findings suggest that many of the numerous attributes

we use to understand, classify, and differentiate objects in the world can be roughly captured in a relatively low-dimensional subspace. Moreover, these findings motivate a culturally and psychologically informed interpretation of distributed representations of language. If many of the key attributes we use in making distinctions and decisions in daily life are well described as a linear combination of a few latent attributes, then these features may be entangled in a language model's representation space, not out of an accident of superposition, but by consequence of their natural correlation in language and thought.

## 3 DATA AND METHODS

Our analyses draw upon three forms of data: (i) human ratings of words on semantic scales collected with a survey, (ii) projections of tokens in LLM embedding spaces corresponding to the same set of words and semantic scales, and (iii) LLM next-token probabilities for a task resembling the survey task. We run tests on models from the instruction-tuned Gemma-3 family (1B, 12B, and 27B). Results in the Appendix show similar patterns for Llama-3 and Qwen-2.5 models with sizes up to 72B parameters (albeit with moderately weaker relationships).

### 3.1 SURVEY DATA

We use data from a recent survey of semantic associations collected by Boutyline and Johnston Boutyline & Johnston (2025). The survey was fielded through the Prolific platform to an online quota sample of 1,750 respondents approximately representative of the US population along major socio-demographic categories. The survey has respondents rate 360 words along a set of 40 semantic scales. To minimize burnout, each respondent was presented with a subset of all word/scale pairs; each word was ultimately rated on each scale by an average of 24 respondents. For our analyses, we drop 59 words that do not correspond to single tokens in our LLM embedding spaces. We also exclude from analysis 12 scales because they are too similar to other scales, (e.g. *human-machine* with *natural-artificial*), or because variance was too high for robust estimates (e.g. *liberal-conservative*), or due to an insufficient number of relevant antonym pairs in the embedding space (e.g. *deserving-undeserving*). The resulting analytic sample is 301 words rated on 28 distinct semantic scales. More information about the survey and our analytic sample are presented in the Appendix.

### 3.2 MEASURING FEATURE DIRECTIONS

To extract feature directions from LLM embeddings, we apply a method originally developed for word embedding models Caliskan et al. (2017); Garg et al. (2018) that has been successfully extended to LLMs Tigges et al. (2023); Arditi et al. (2024). This method formalizes the intuition that features are identifiable as the direction between antonyms in the embedding space, and that semantically similar antonym pairs share similar directional vectors (e.g., *good-bad* has high similarity to *great-awful*). Specifically, given a set of antonym pairs for a feature of interest, we compute the normalized difference between the embedding vectors of each antonym pair and then take the mean of these normalized differences, as shown in Eq. equation 1:

$$\mathbf{d}_f = \frac{1}{N} \sum_{j=1}^{N} \frac{\mathbf{ant}_j^+ - \mathbf{ant}_j^-}{\|\mathbf{ant}_j^+ - \mathbf{ant}_j^-\|} \tag{1}$$

where $\mathbf{d}_f$ is the vector for semantic feature $f$, $N$ is the total number of antonym pairs (we use 10 pairs for each feature), and $\mathbf{ant}_j^+$ and $\mathbf{ant}_j^-$ are embedding word vectors corresponding to the positive and negative antonyms in the $j$-th antonym pair. We restrict our analysis to words that are fully represented in a single token.

We quantify semantic associations in the embedding space as the cosine similarity between the target token and the respective feature vector, which is equivalent to the linear projection of the two normalized vectors. We construct a dataset of projections calculated in this way mirroring the survey data described above. Specifically, we construct 28 feature vectors corresponding to the semantic scales measured in the survey (e.g. kind-cruel, foolish-wise, etc.). Then, for each of the 301 words rated in the survey, we take the associated token vector in the embedding matrix and calculate its

cosine similarity with each of the 28 feature vectors. The resulting dataset provides the normalized linear projection of 301 tokens on 28 features and is directly comparable to the survey dataset.

As a preliminary step, we calculate the correlation between the projections of word vectors onto the extracted semantic feature vectors and the human ratings of those words on the corresponding semantic scales. These correlations serve to validate our method of deriving semantic associations from LLM embeddings. We compare our approach to an alternate method proposed by Park et al. Park et al. (2023). They apply a whitening operation to the model's unembedding space prior to identifying feature directions and performing interventions to isolate "causally separable concepts" and minimize off-target effects. Specifically, they left-multiply their unembedding vectors by the inverse square root of the covariance matrix of the unembedding space ($\mathbf{\Sigma}^{-1/2}$), orthogonalizing the axes with the greatest variance across tokens. Although effective at reducing off-target effects, we hypothesize that this transformation may distort feature representations by artificially separating semantically related features.

After validating our projections as indicators of semantic association akin to survey ratings, we then subject both the survey and projection datasets to a series of analyses to compare their correlational structures. We first calculate a complete correlation matrix between all 28 semantic scales measured in the survey across all words in our analytic sample. We do the same with the word vector projections on the semantic feature vectors, producing a comparable set of correlation matrices. We also calculate the cosine similarity between each pair of semantic feature vectors in the LLM embeddings. While the correlations between projections reveals whether data points are correlated on these axes, the cosine similarities reveal the extent to which the axes themselves are non-orthogonal.

Finally, for both the survey and projection datasets, we apply PCA and examine the variance explained by each component and the variables loading highest on each of the first three components. These parallel analyses serve to compare the correlational structure of the 28 semantic scales measured with survey data and with embedding projections to identify common patterning.

### 3.3 INTERVENTIONS AND OFF-TARGET EFFECTS

After assessing correlations between features, we evaluate causal relationships by measuring the impact of interventions on the embedding space on next-token probabilities. For each token $i$, each antonym pair $j$, and each semantic feature $f$, we construct prompts using the following template:

> **USER:** Do you associate {$token_i$} more with {$first\_antonym_{f,j}$} or {$second\_antonym_{f,j}$}? Please select one of these two words with no formatting.
> **ASSISTANT:** Between {$first\_antonym_{f,j}$} or {$second\_antonym_{f,j}$}, I think {$token_i$} is more

We then calculate the probabilities of each of the two antonyms (e.g. "kind" and "cruel") as the next token, then normalize the probabilities so they sum to 1. We repeat this process across all ten pairs of antonyms for each semantic feature $f$ and across both orderings of the two antonyms to improve robustness, and average the resulting normalized probabilities, as described in Eq. equation 2.

$$p_{\text{norm}}(ant_f^+|w_i) = \frac{1}{N} \sum_{j=1}^{N} \frac{p(ant_{f,j}^+|w_i)}{p(ant_{f,j}^+|w_i) + p(ant_{f,j}^-|w_i)} \tag{2}$$

In Equation 2, $p_{\text{norm}}(ant_f^+|w_i)$ represents the mean normalized probability of selecting the positive antonyms associated with feature $f$, given the target token $w_i$. The terms $ant_{f,j}^+$ and $ant_{f,j}^-$ refer to the positive and negative antonyms for the $j$-th antonym pair associated with feature $f$, and $N$ denotes the total number of antonym pairs considered for that feature.

After collecting baseline probabilities, we perform token-level interventions for each prompt, nudging the target token in the direction of each of the 28 feature vectors. Specifically, we calculate the 28 feature vectors according to the procedure outlined above (see Eq. equation 1), then for each token $i$, we modify that token's representation in the embedding matrix by moving it in the direction of semantic feature $\mathbf{d}_f$. This is achieved by adding the scaled semantic feature vector $\mathbf{cd}_f$ to the

target token $\mathbf{w}_i$. The intervention vector is scaled to have a magnitude of $\pm 0.35 \cdot \|\mathbf{w}_i\|$; preliminary tests indicated that this level of intervention preserves output coherence and token definitional meaning. After applying the intervention, we normalize the modified token vector to restore its original magnitude $\|\mathbf{w}_i\|$.

# 4 SEMANTIC STRUCTURE IN SURVEYS AND EMBEDDINGS

## 4.1 PROJECTION DATA

We begin by assessing the correlation between human semantic ratings and the projections of words on the corresponding feature vectors in LLM embeddings (Figure 1). We see that across features, the correlations between embedding projections and human ratings are high, ranging from 0.3 to 0.7. Given the stark differences between these approaches for estimating semantic association and the high level of measurement error inherent to measuring subjective phenomena like semantic ratings, these values indicate a strong correspondence between psychological associations and the positioning of tokens in LLM embedding spaces.

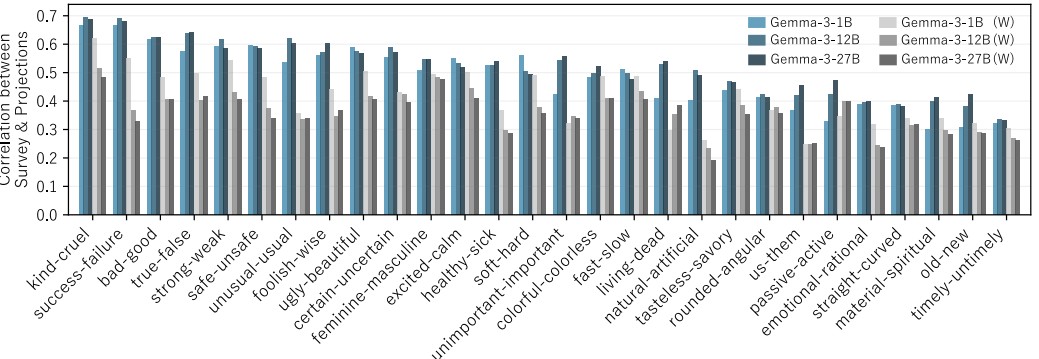

Figure 1: Pearson correlations between human ratings and token embedding projections for 301 words on 28 semantic features. Correlations with "whitened" embedding space are labeled (W).

Applying the whitening transformation described by Park et al. Park et al. (2023) to the embeddings reduces the average token-level correlation between our 28 projection scores and the corresponding survey ratings by approximately 20%. The decline is consistent with our hypothesis: whitening makes the token cloud isotropic and de-correlates directions that were previously allowed to share variance; human evaluative judgments, however, draw on that shared variance, and thus removing these overlaps reduces the representation's fidelity to psychological and cultural associations. This finding suggests that the mild non-orthogonality present in embeddings is a faithful reflection of semantic structure rather than noise.

Next, we compare the semantic structure as derived from survey ratings to values extracted from an LLM embedding. Panel A of Figure 2, as anticipated, shows high correlations between semantic scales in the survey data. Panels B through D display associations between features in the Gemma-3-12B embedding (results for other LLMs are included in the Appendix). The correlations between the LLM features are not as high as those observed in the survey data, but are still substantial, both when the vocabulary is restricted to the survey items (Panel B) and for the entire embedding embedding vocabulary (Panel D). These correlations appear especially strong when considered against the alternative hypothesis that models seek to preserve orthogonality between features. Moreover, the structure of correlations in the projection data is highly similar to that of the survey data. We take the entries of the survey correlation matrix and the corresponding entries of the projection correlation matrix, and find that their Pearson correlation is 0.82 when restricted to the survey vocabulary and 0.73 for the full vocabulary. However, even for the full vocabulary, it is possible that we are still observing an artifact of observed data points rather than an alignment of the latent features themselves. To assess the alignment of the semantic features themselves, we present the distribution of cross-feature cosine similarities in Panel C of Figure 2 . While many features are nearly orthogonal, with cosine similarities less than 0.05, a large share display substantial similarities, with

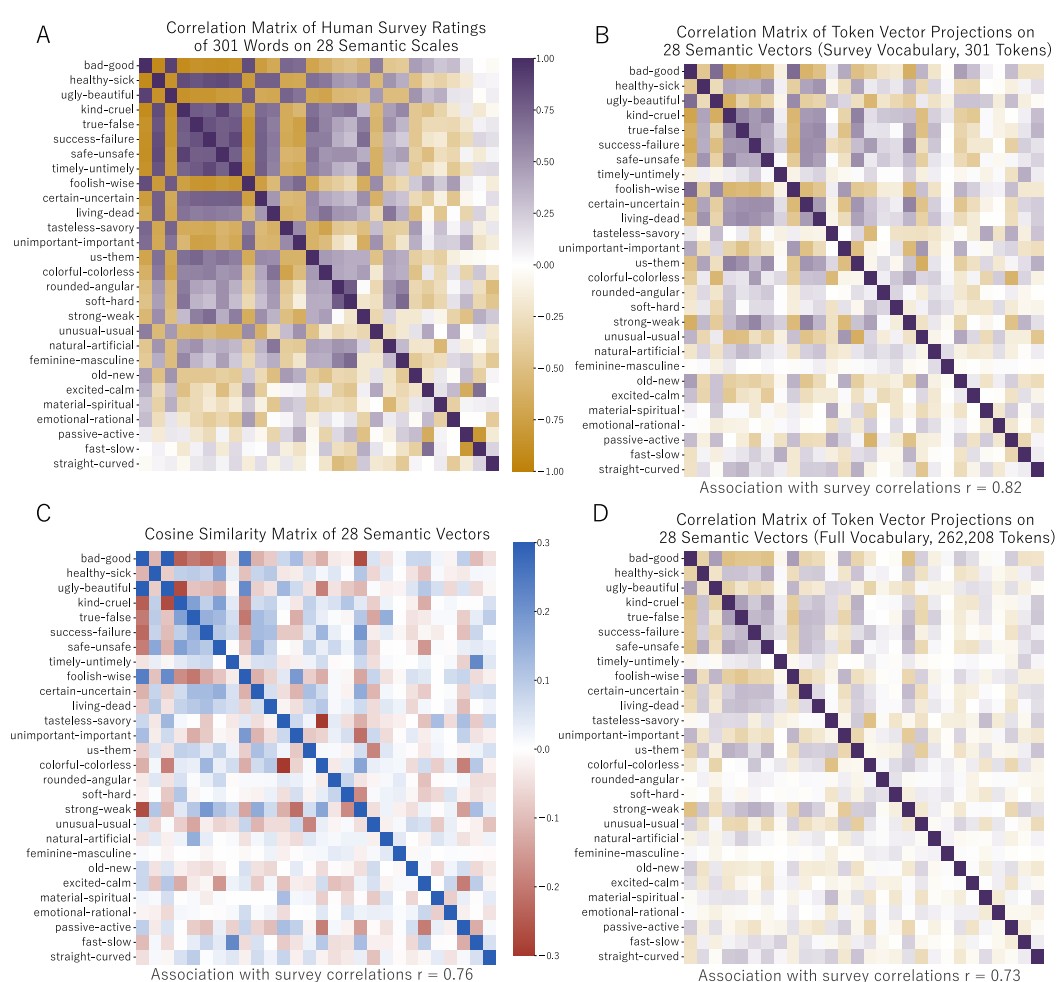

Figure 2: Panel (A): Pearson correlations between human ratings of 301 words on 28 semantic scales. Panel (B): Pearson correlations between projections of 301 word vectors (survey vocabulary) on 28 semantic feature vectors, Gemma-3-12B embedding. Panel (C): Cosine similarities between pairs of semantic feature vectors in the Gemma 3-12B embedding. Panel (D): Pearson correlations between projections of all token vectors on 28 semantic feature vectors, Gemma-3-12B embedding.

many between 0.1 and 0.3. The correlation between semantic vector cosine similarities and the associated correlation coefficient from the survey dataset is 0.76, strongly suggesting that humanly meaningful semantic associations are encoded in the geometric relations between feature vectors in the embedding space.

## 4.2 MEASURING SEMANTIC STRUCTURE

Next, we examine results from principal components analysis (PCA), presented in Figure 3, to assess whether correlations from survey ratings and from LLM projections are reducible to similar latent covariance structures. Panel A displays the variance explained by each of the top components in the survey data and each of the three LLM embeddings. While a larger share of variance is explained by the first three components in the survey data, the first three components capture a substantial amount of variance for the LLM embeddings as well, especially against the baseline of perfect orthogonality of features, in which each of the 28 components would only explain 3.6% of the variance.

Examining the loadings on the top three components, we see that the survey cleanly reproduces the Evaluation, Potency, and Activity dimensions. The first component is most closely related to

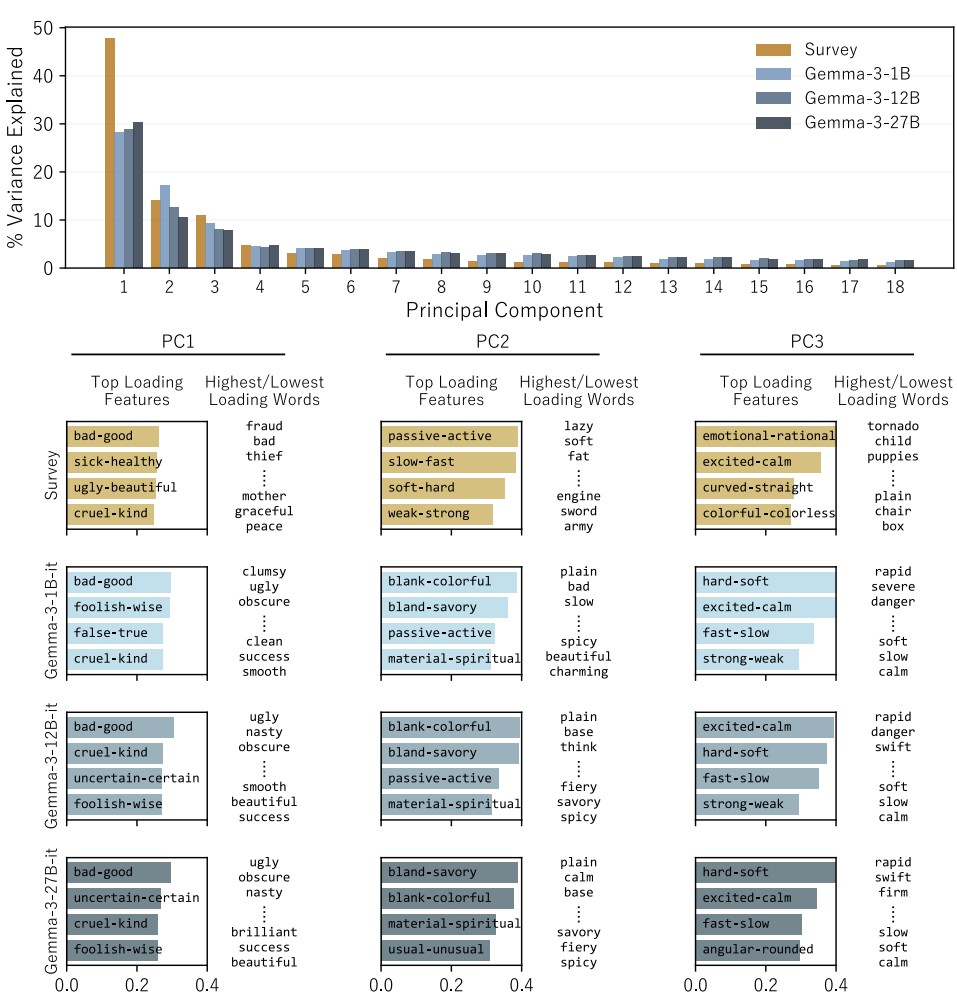

Figure 3: Results from principal components analysis applied to survey ratings of 301 words on 28 semantic scales and the projections of those 301 word vectors on 28 corresponding semantic feature vectors in the embeddings of Gemma 3-1B, 12B, and 27B. Panel (a): Scree plot of variance explained by each of the top 18 components. Panel (b): Highest loading features on the first, second, and third principal components, beside highest and lowest scoring words on each component.

*bad-good* and *sick-healthy*, and the most extreme values are found in words like "fraud" and "thief" on one end and "graceful" and "peace" on the other. At first glance, the second component appears related to Activity, with *passive-active* and *slow-fast* being the highest loadings, but the extreme values of "lazy" and "soft" versus "sword" and "army" suggest weakness versus power. By contrast, the most extreme words on component 3, "tornado," "child," and "puppies" on one end and "plain," "chair," and "box" on the other, more clearly indicate activity and motion versus inanimate stillness.

The Gemma models of all three sizes display similar results. As in the survey, the first component clearly captures Evaluation, with *bad-good* emerging as component 1's highest loading feature across all models. The second component admittedly does not correspond to Potency so much as "vibrancy." The highest loading features are *blank-colorful* and *bland-savory* across model sizes, and the highest loading words are "plain," "slow," and "base" on one end and "spicy," "fiery," and "charming" on the other. The third component corresponds approximately to Activity, with *excited-calm* and *fast-slow* consistently appearing among the top loading features, and words like "rapid" and "danger" versus "calm" and "slow" exhibiting the most extreme values. Overall, these results suggest that projections along the 28 semantic feature vectors reduces to a low-dimensional subspace not identical, but closely resembling the one produced from survey responses.

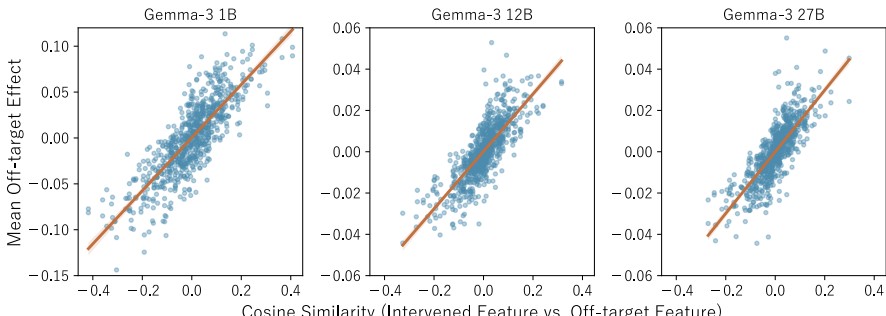

Figure 4: Average size of off-target effects by cosine similarity of target and off-target semantic feature vectors in Gemma 3 models. Effects are measured as change in normalized probability of the next token being antonym 1 vs. antonym 2.

## 5 PREDICTING OFF-TARGET EFFECTS

The previous analyses establish meaningful correlations between semantic features in LLM embeddings resembling structures found in human survey data. However, the implications of this feature "entanglement" is unclear. In this final stage of analysis, we explore whether position on one feature vector is causally related to the model's evaluations of word's associations on off-target but correlated semantic features. Results are presented in Figure 4. Each point represents a pair of two feature vectors, with their cosine similarity of the feature pair on the x-axis and the magnitude of the off-target effect on the y-axis. We note a strong linear association between the "entanglement" of feature vectors and the size of their off-target effects. While effects are strongest in smallest model, with the most aligned feature pairs exhibiting average off-target effects of 10 percentage points, the 12B and 27B models also display a clear association and substantial off-target effects. These results suggest that the position of words along feature vectors predictably influences how the model judges them semantically. Moreover, alignment between many feature vectors is strong enough that substantial off-target effects predictably occur.

## 6 CONCLUSION

LLMs are notable for the high dimensionality of their representations, and their performance scales consistently as a power-law function of parameter counts over many orders of magnitude Kaplan et al. (2020). This phenomenon seems to imply that building an effective world model requires an enormous number of dimensions. However, prior psychological research suggests that many of the seemingly diverse axes of meaning that we use to communicate and understand the world are roughly represented as a linear combination of just three latent factors Osgood et al. (1957); Heise (2010). Our findings suggest that, as with humans, the representation of semantic associations is relatively low-dimensional in LLM embeddings. This finding appears encouraging for technical AI safety efforts, because it suggests that many relevant semantic features (e.g. kind-cruel, safe-unsafe, us-them) may share a common low dimensional subspace, simplifying the tasks of identification, auditing, and control Zou et al. (2023).

Furthermore, this study goes beyond most existing research on feature identification by explicitly measuring and modeling the relations between multiple features. While identifying individual feature vectors remains important for purposes of model auditing and steering, a more complete understanding of LLMs' internal operations must account for how features are positioned in relation to one another and how this co-location shapes model behavior. Yet this study provides only the first steps toward mapping the geometry of feature space. Future studies will need to consider how these geometries are transformed over the course of the model's layers and components, and how the relations between features in activation space may be dynamically reconfigured by the preceding tokens in the context sequence. Humans understand the world not as a set of isolated features, but through relations, commonalities, and differences. Advancing our understanding of how LLMs make sense of the world may require not just assembling features into circuits, but into conceptual maps.

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

# A   APPENDIX

## A.1   SURVEY DESCRIPTION

The survey we analyze, fielded by Boutyline and Johnston Boutyline & Johnston (2025) in 2024, is an update of a classic semantic differential survey by Jenkins et al. Jenkins et al. (1958) administered in the 1958. The authors of the original survey asserted that randomly sampling from "all concepts" in English is impossible, and therefore they intentionally selected words from the following classes: (i) the Kent-Rosanoff Association Test, (ii) words that scored high on a previous scale of "meaningfulness", (iii) words that "sample the space" by maximizing semantic difference, (iv) words from previous semantic differential studies, and (v) miscellaneous words spanning social, political, economic, and clinical domains. The authors of the modern replication pre-tested the word list to make sure that the words are still familiar to modern-day respondents, and replaced four words with better known synonyms. We include in our sample all words in Boutyline and Johnston's survey that are represented with a single token, resulting in an analytic sample of 301 words.

Files containing the full list of words and the full sets of antonym pairs used to estimate each semantic feature vector are included in the Supplementary Materials.

## A.2   EMBEDDING DETAILS

| Model | Dimensions | Embeddings |
|---|---|---|
| Gemma-3-1B | 1152 | tied |
| Gemma-3-12B | 3840 | tied |
| Gemma-3-27B | 5376 | tied |
| Llama-3.2-3B | 3072 | tied |
| Llama-3.1-8B | 4096 | untied |
| Llama-3.1-70B | 8192 | untied |
| Qwen-2.5-1.5B | 1536 | tied |
| Qwen-2.5-7B | 3584 | untied |
| Qwen-2.5-72B | 8192 | untied |

Table 1: Dimensionality and whether embeddings and unembeddings are tied for analyzed models.

## A.3 SUPPLEMENTARY ANALYSES

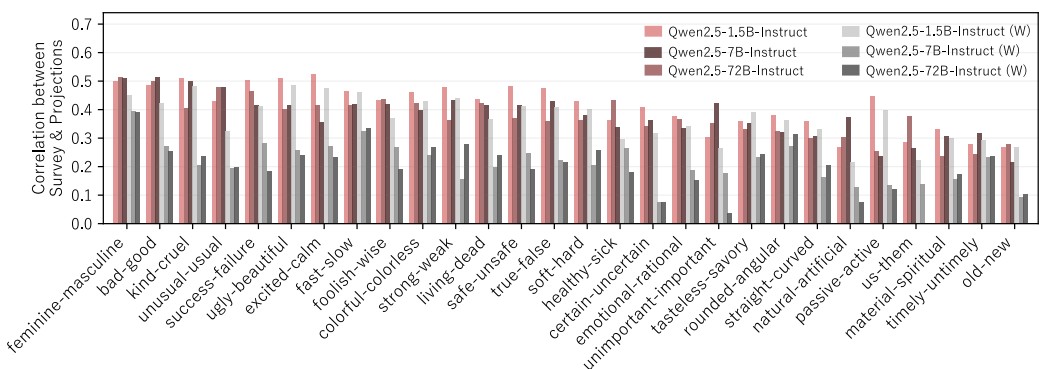

Figure 5: Pearson correlations between human ratings and token embedding projections for 301 words on 28 semantic features in Qwen2.5 family models. Correlations using the "whitened" embedding space are labeled with (W).

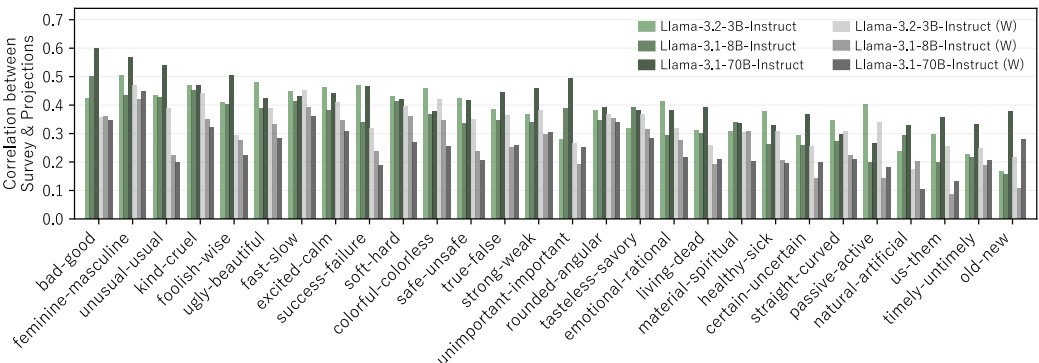

Figure 6: Pearson correlations between human ratings and token embedding projections for 301 words on 28 semantic features in Llama 3 family models. Correlations using the "whitened" embedding space are labeled with (W).

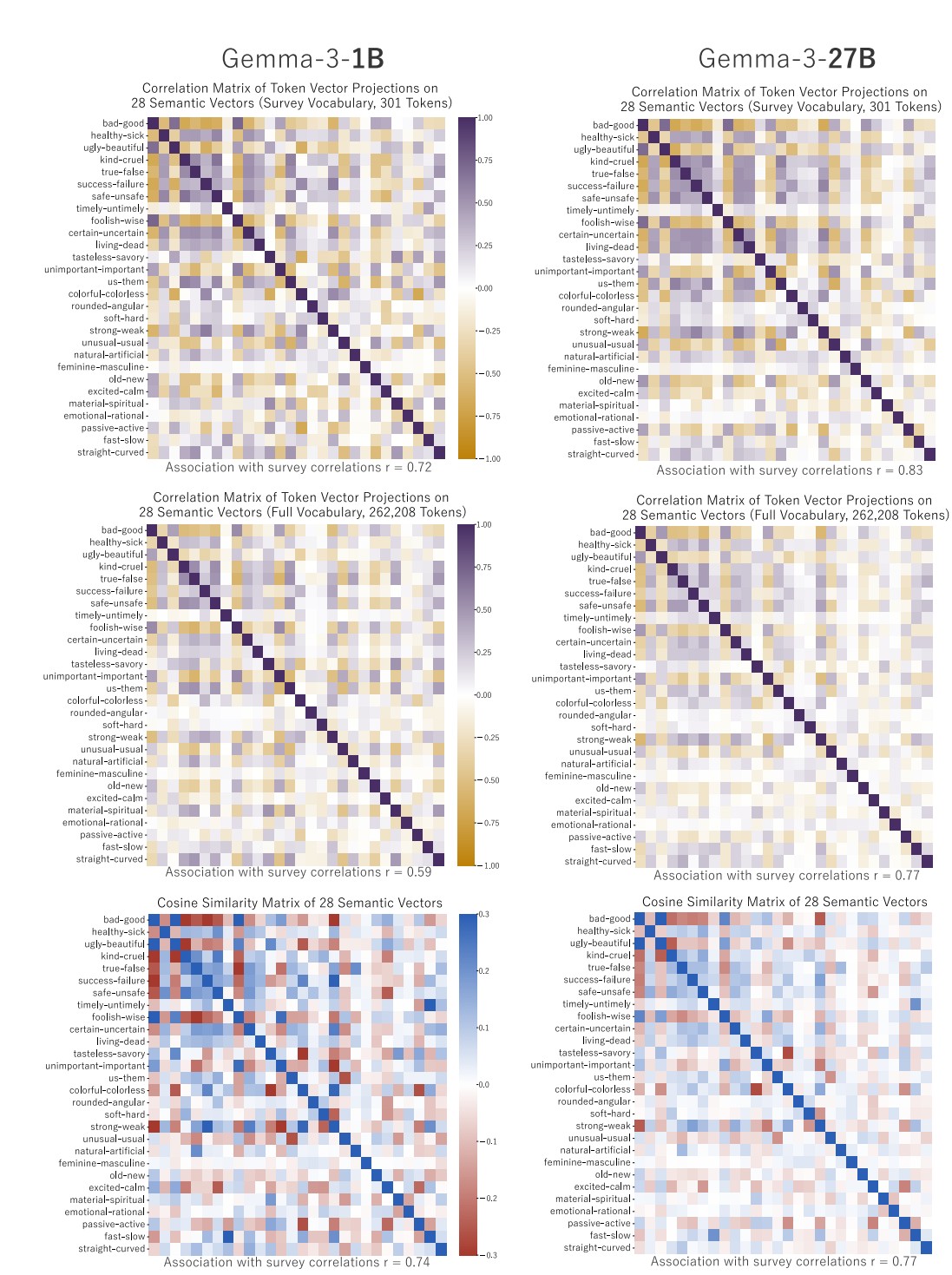

Figure 7: Results from Gemma-3-1B and Gemma-3-27B: Pearson correlations between projections of 301 word vectors (survey vocabulary) on 28 semantic feature vectors, Pearson correlations between projections of all token vectors on 28 semantic feature vectors, and cosine similarities between pairs of semantic feature vectors.

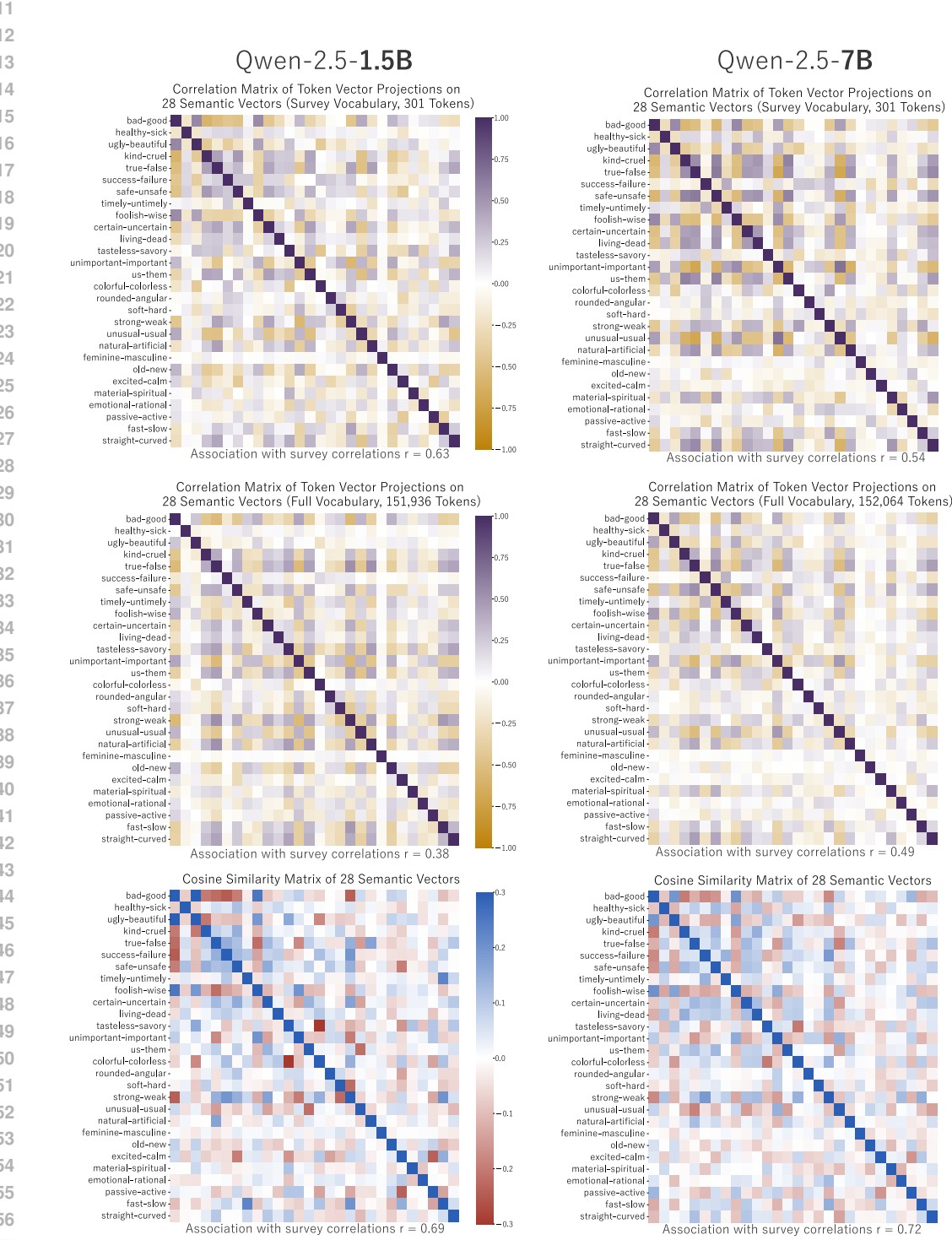

Figure 8: Results from Qwen-2.5-1.5B and Qwen-2.5-7B: Pearson correlations between projections of 301 word vectors (survey vocabulary) on 28 semantic feature vectors, Pearson correlations between projections of all token vectors on 28 semantic feature vectors, and cosine similarities between pairs of semantic feature vectors.

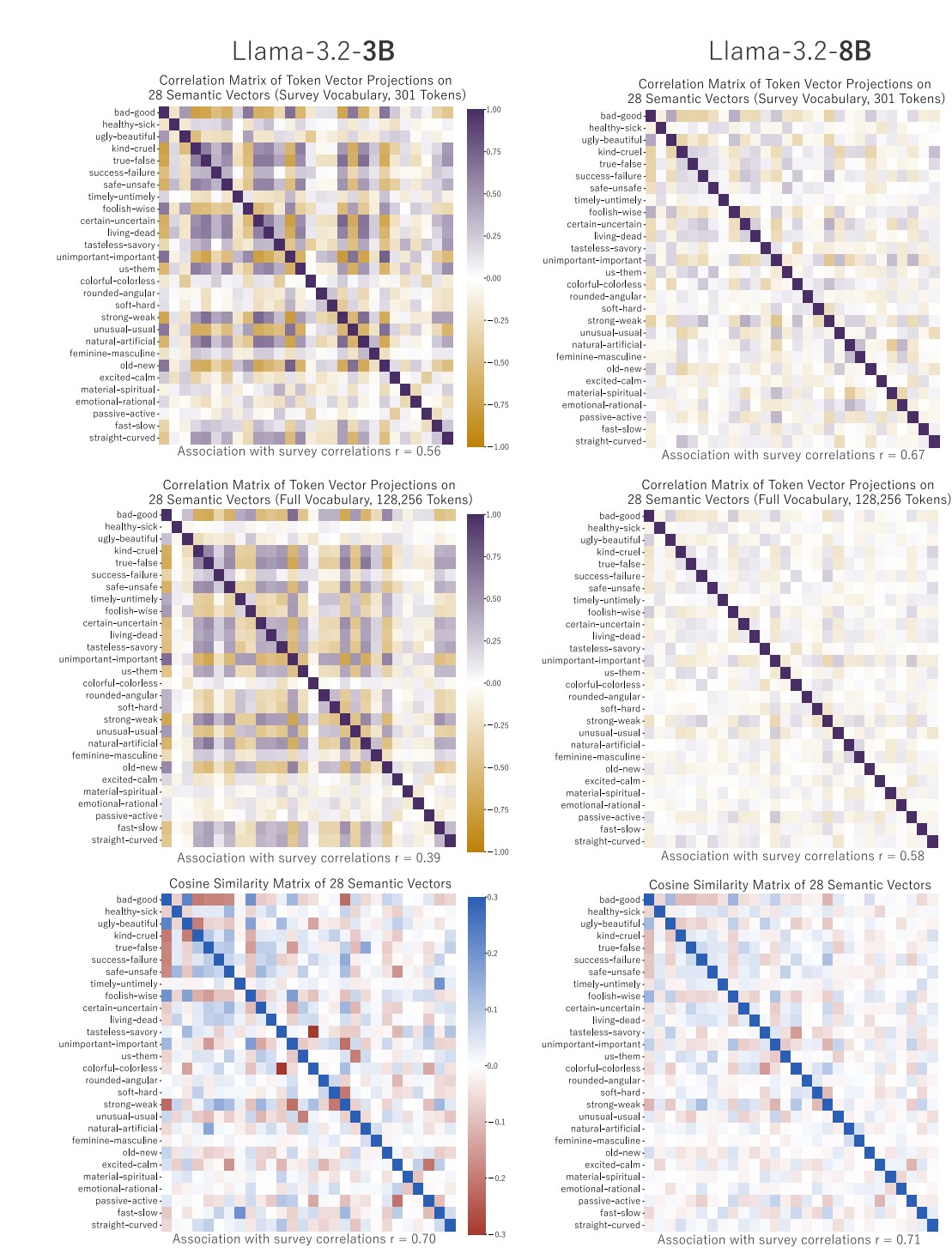

Figure 9: Results from Llama-3.2-3B and Llama-3.1-8B: Pearson correlations between projections of 301 word vectors (survey vocabulary) on 28 semantic feature vectors, Pearson correlations between projections of all token vectors on 28 semantic feature vectors, and cosine similarities between pairs of semantic feature vectors.

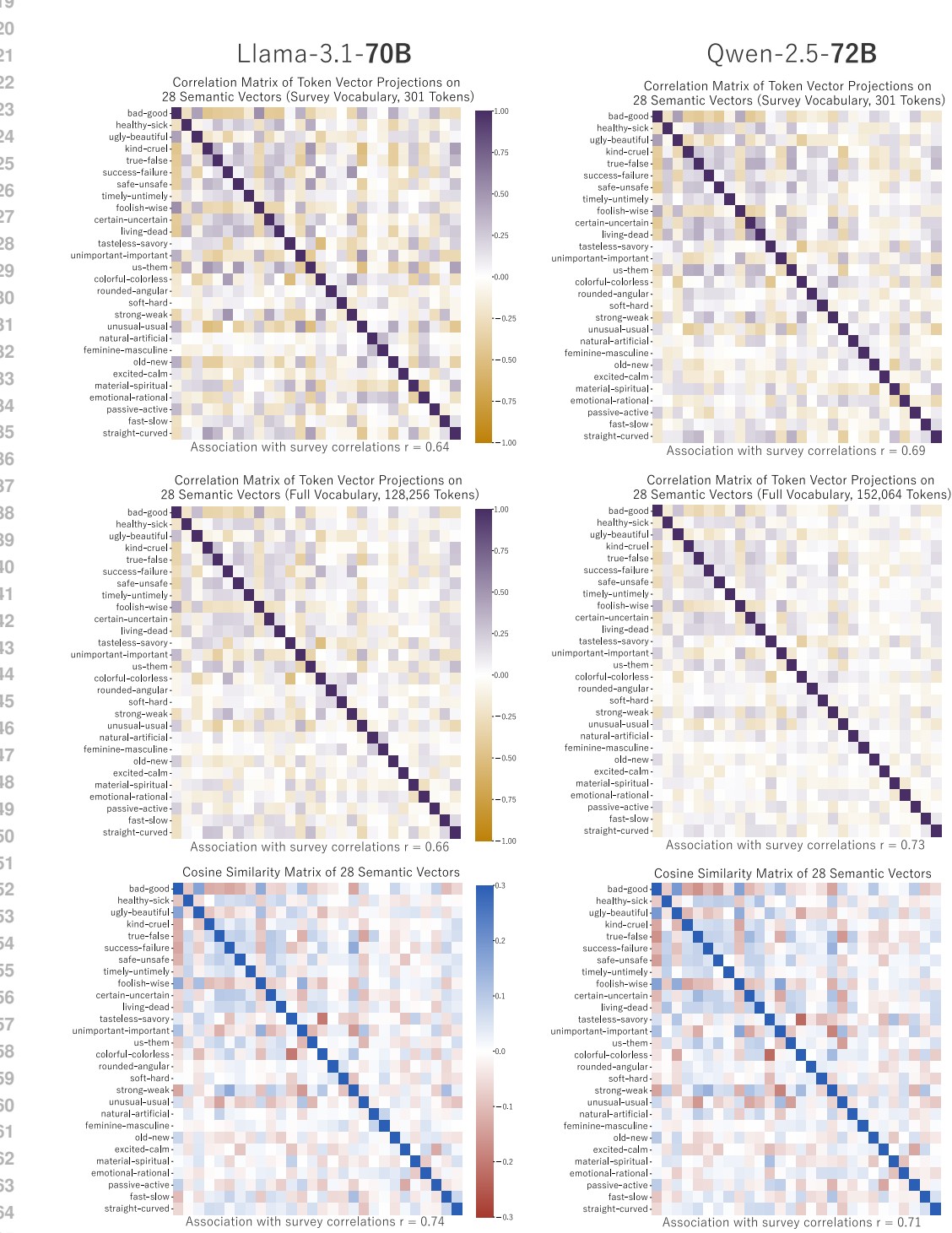

Figure 10: Results from Llama-3.1-70B and Qwen-2.5-72B: Pearson correlations between projections of 301 word vectors (survey vocabulary) on 28 semantic feature vectors, Pearson correlations between projections of all token vectors on 28 semantic feature vectors, and cosine similarities between pairs of semantic feature vectors.

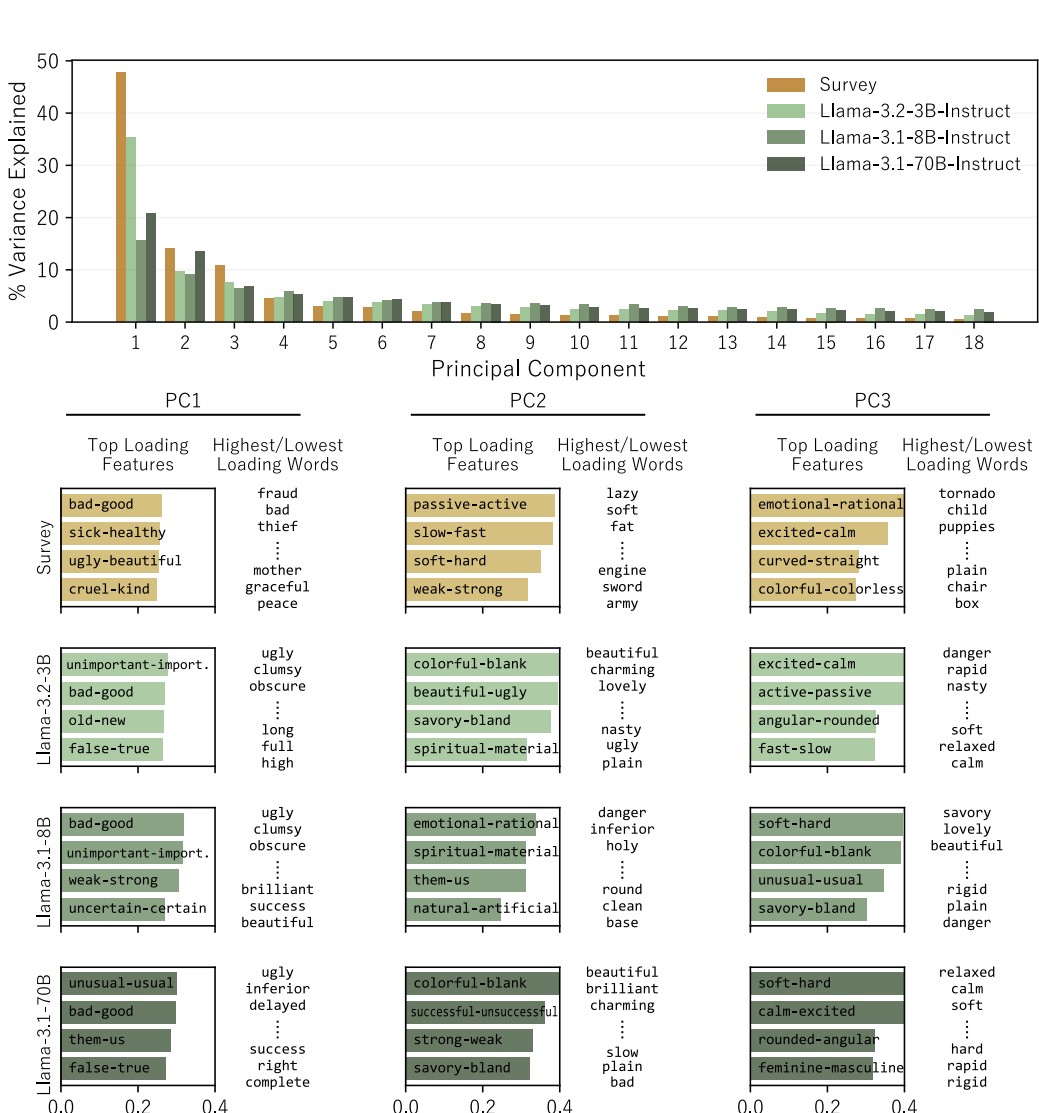

Figure 11: Results from principal components analysis applied to survey ratings of 301 words on 28 semantic scales and the projections of those 301 word vectors on 28 corresponding semantic feature vectors in the embeddings of Llama-3 3B, 8B, and 70B models. Panel (a): Scree plot of variance explained by each of the top 18 components. Panel (b): Highest loading features on the first, second, and third principal components, beside highest and lowest scoring words on each component.

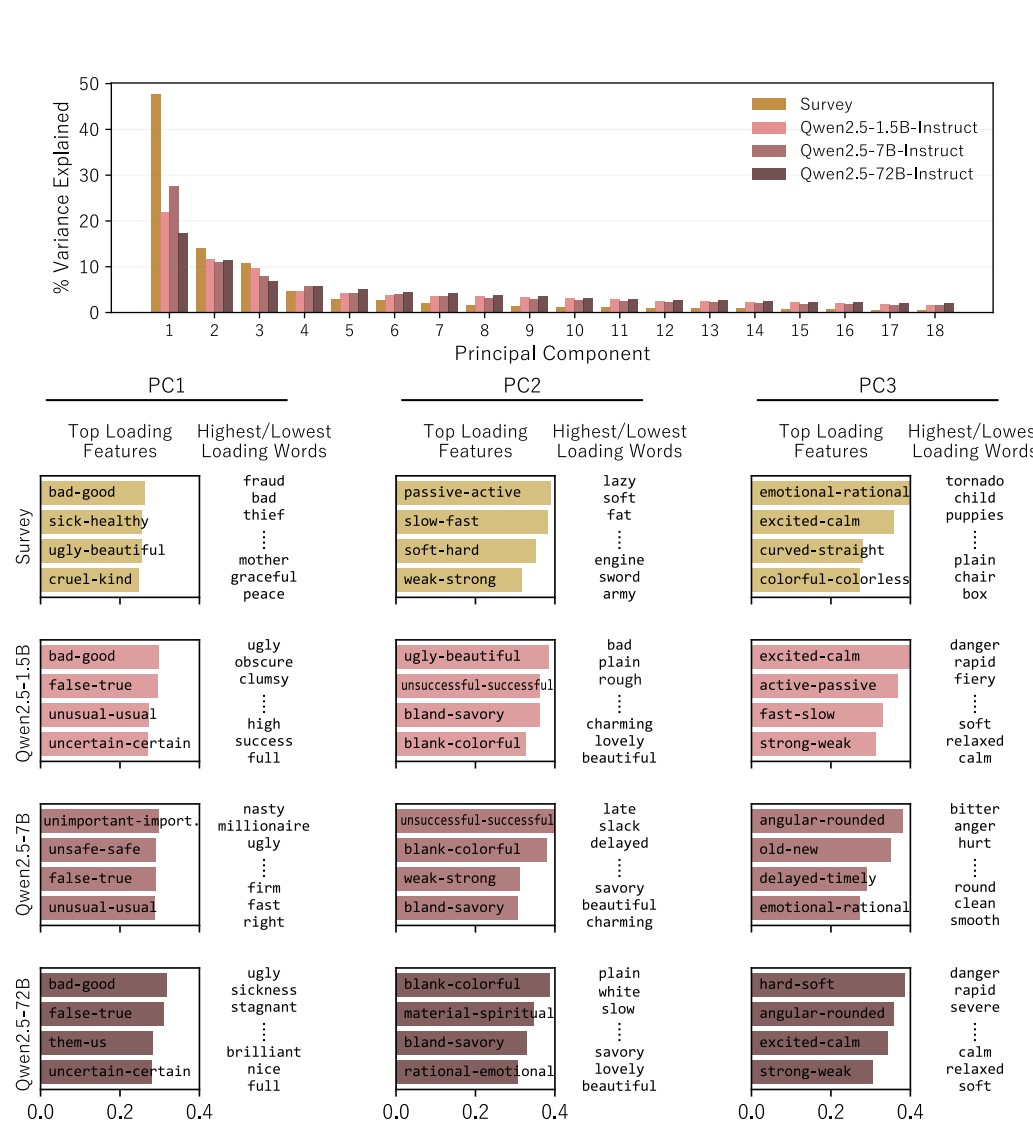

Figure 12: Results from principal components analysis applied to survey ratings of 301 words on 28 semantic scales and the projections of those 301 word vectors on 28 corresponding semantic feature vectors in the embeddings of Qwen2.5 1.5B, 7B, and 72B models. Panel (a): Scree plot of variance explained by each of the top 18 components. Panel (b): Highest loading features on the first, second, and third principal components, beside highest and lowest scoring words on each component.

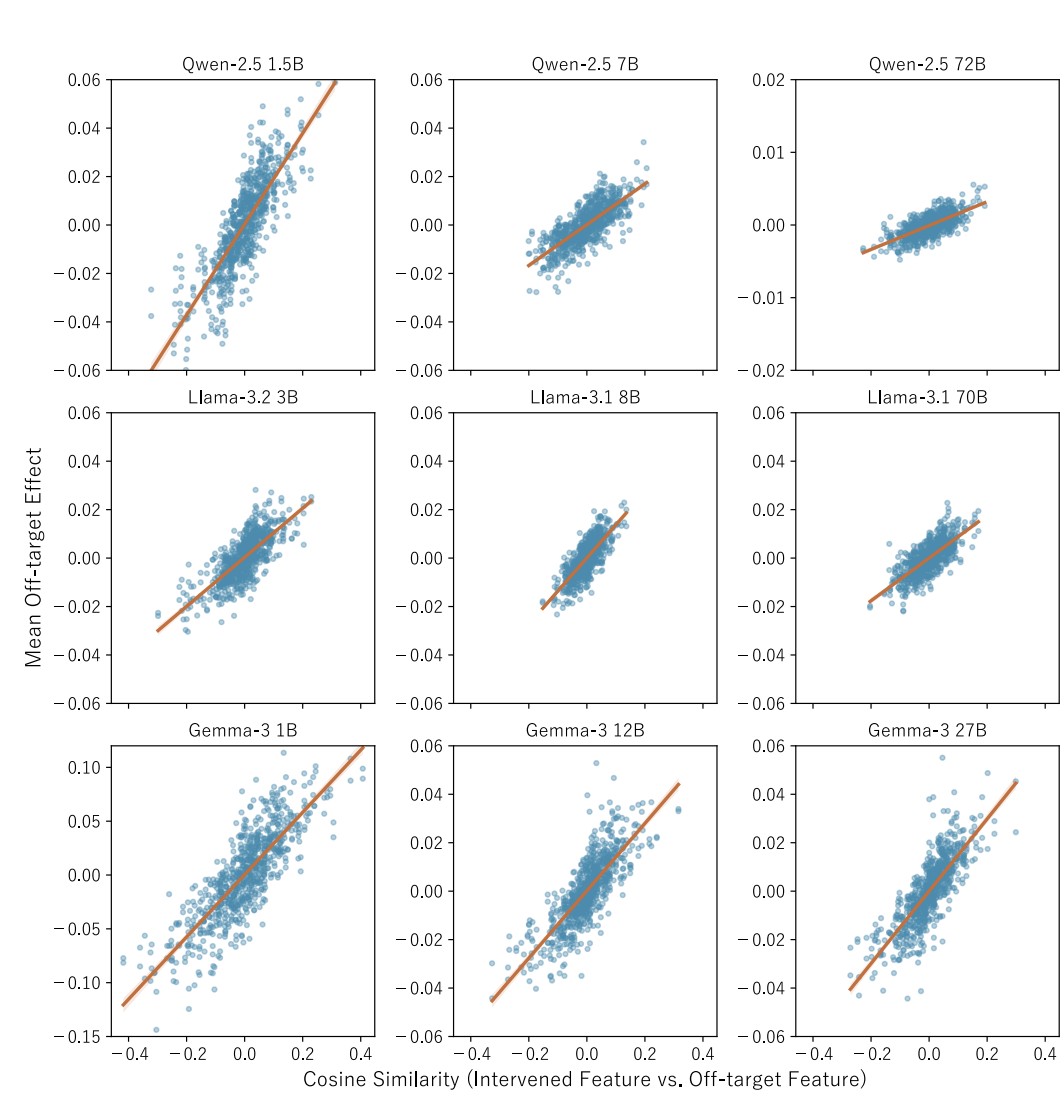

Figure 13: Average size of off-target effects by cosine similarity of target and off-target semantic feature vectors in Qwen, Llama, and Gemma models. Effects are measured as change in normalized probability of the next token being antonym 1 vs. antonym 2.

