# OpenReview forum: "Semantic Structure in Large Language Model Embeddings"
_ICLR.cc/2026/Conference — Submitted to ICLR 2026_

### Official Review · Reviewer_3DvR · 2025-10-24

**Soundness:** 3
**Presentation:** 3
**Contribution:** 2
**Rating:** 6
**Confidence:** 3

**Summary:**

This paper explores the semantic organization of token embeddings in large language models (LLMs) through a cognitive science perspective. The authors present an interesting analysis revealing two main findings: (1) token embeddings show alignment with human semantic ratings, and (2) there is a correlation between embedding features and off-target effects in embedding steering.

**Strengths:**

The paper is well written and clearly structured. The experimental design is solid and supports the authors’ claims effectively. The findings are interesting and open promising research directions at the intersection of cognitive science and large language models.

**Weaknesses:**

The visualizations could be clearer. For example, the correlation matrices presented in the appendix are difficult to interpret and compare in their current form.

Moreover, the paper’s practical impact appears limited. While the results are conceptually interesting, it remains unclear how these insights could be applied to improve LLM architectures or their downstream performance.

**Questions:**

The authors write:

> Whitening makes the token cloud isotropic and de-correlates directions that were previously allowed to share variance; human evaluative judgments, however, draw on that shared variance, and thus removing these overlaps reduces the representation’s fidelity to psychological and cultural associations.

While this interpretation is supported by the results, whitening might also have a regularization effect that improves performance in LLMs. Without targeted experiments, it is difficult to disentangle these two effects. It would be helpful to acknowledge this possibility in the paper.

Additionally, in Figure 2 and throughout the appendix, the correlation matrices comparing token projections across various models are hard to visually compare. It would strengthen the analysis to include a single aggregate metric summarizing how well these correlations align with human semantic data. A comparison across embedding dimensionality or model size would also provide further value to the analysis.

---

> ### Author Response · Authors · 2025-11-23
>
> Thank you for these comments.
>
> Given your response as well as Reviewer 1Pp729’s comments, we have decided to replace the heatmaps in Figure 2 and instead represent the data with scatterplots in which the x-axis is a feature-pair’s correlation in the survey data and the y-axis is the feature-pairs correlation among token projections (or the cosine similarity between the feature pair). The similarity between survey results and embedding results can then be clearly perceived as the correlation of the datapoints in a scatterplot rather than by eyeballing two heatmaps.
>
> You mention that “it would strengthen the analysis to include a single aggregate metric summarizing how well these correlations align with human semantic data.” We do present below each panel in Figure 2 the Pearson correlations between the values in the survey heatmaps and the values in each other heatmap, but these values (0.82, 0.76, and 0.73) will be more clearly represented in a scatterplot.
>
> You express concern that the practical impact is limited. We admit that the primary contribution of this paper is scientific rather than directly practical. Its objective is to improve our understanding of how concepts are geometrically represented in LLM latent spaces. This work does have practical implications, though. Our findings regarding off-target effects could be useful to practitioners who are using “steering vectors.” Specifically, our work shows that off-target effects are not merely random, but the geometric alignment of semantically related features leads to an increased probability of off-target spillovers.
>
> Finally, you suggest that whitening may improve LLM performance through regularization. We think this is an intriguing possibility and do not disagree. But once again, the purpose of this study is not to discover ways to improve LLMs, but to understand how they internally represent concepts. One plausible hypothesis is that LLMs try to keep features orthogonal to minimize interference, but some geometric alignment is a regrettable byproduct of packing >>n features in an n-dimensional space. Our evidence suggests that the alignment between features is not a random byproduct of compression, but encodes meaningful semantic relations (just like in word2vec). Perhaps this “strategy" of representing features non-orthogonally has downsides as well, as you suggest. However, it is not our intention to argue that this approach to representation is optimal, but merely to show that this is what LLMs are doing by default.

---

### Official Review · Reviewer_1Pp7 · 2025-10-29

**Soundness:** 2
**Presentation:** 3
**Contribution:** 1
**Rating:** 2
**Confidence:** 4

**Summary:**

The paper empirically verifies that the spatial arrangement of LLM embeddings aligns with low-dimensional psychological models of meaning, and demonstrates that the geometric alignment (non-orthogonality) of semantic features predicts intervention side effects.

**Strengths:**

The paper is highly readable and the topic of understanding how LLMs represent meaning, particularly in relation to human cognitive models (Evaluation, Potency, Activity), is of broad interest.

**Weaknesses:**

The paper is insufficiently substantive for a top-tier machine learning conference. It offers no new theoretical insight into how semantics are encoded but provides additional empirical evidence for an existing, well-known principle that the geometric arrangement of embeddings (angles and vector addition) encodes semantic meaning.

The central finding that feature "entanglement" leads to predictable off-target effects is fundamentally the expected mathematical consequence of performing linear manipulations in a non-orthogonal feature space. While the authors claim previous studies treated these effects as random, the mechanism confirms the precise problem that prior work in causal interpretability (e.g., Park et al.) sought to solve by defining causally separable concepts.

Furthermore, the paper’s argument against minimizing feature correlation via techniques like the whitening operation constitutes a potentially misdirected comparison (a "straw-man argument"), as related literature on feature steering aims to orthogonalize causally separable concepts, not necessarily every possible semantic dimension en masse. Thus naively orthogonalising is perhaps not a good idea and it would be of interest to consider which concepts can/can't be orthogonalised.

The conclusion that "the representation of semantic associations is relatively low-dimensional in LLM embeddings" seems an unsubstantiated generalization. This dimensionality reduction analysis (PCA) was performed only on 28 predetermined scales across a restricted sample of 301 words (tokens), making extrapolation to the full, high-dimensional LLM semantic space unwarranted.

**Detailed points**:
 - referencing is incorrect: use "citep" so that references are in brackets (see sample ICLR paper)
 - the paper would be clearer if it explicitly defined the concept of "almost orthogonal" vectors.
 - the mechanisms discussed (e.g., superposition at line 110) should include a clearer explanation of why a space of dimension n can represent ≫n "nearly orthogonal" vectors.
 - 132: the reason word2vec received attention was also (if not more) due to analogy solvin by vector addition, not only "angles between features".
 - 199/208: The origin and grouping of the antonym pairs used to derive the 28 semantic axes are unclear.
 - 229-240: unclear - is this section a segue/intro to the experiments below, if so this could be more clear and give section references
 - 247-251: this is confusing. If the probability over the "next token" is taken (253), why the "Assistant" part of the prompt?
 - 270: it would be clearer and more reproducible to give an explicit formula for the intervention vector etc.
 - 302: Park et al don't argue for orthogonalising arbitrary concepts (as above) so, this comparison is not necessarily meaningful.
 - Panels A, B, D: it is unclear what to take from this or the logic of the argument. Why is it good/bad that word embeddings are partly aligned with several reference vectors?
 - Panel C: the scale perhaps makes this misleading. Presumably the diagonals have value 1? which is off the (-0.3, 0.3) scale? All values are relatively small (relative to the diagonal) and it would be clearer if the scale reflected this clearly. Also, there doesn't seem any accounting for the phenomenon that in high dimensions, random vectors are more likely to be orthogonal, so orthogonality itself is less "surprising"/meaningful.
 - 363: a histogram of values would be more clear than the heatmap (but, as above, perhaps with reference to the distribution one would expect for random vectors).
 - Section 5: this section, describing the intervention experiment, is concise but unclear and could be better placed or clarified to enhance readability.

**Questions:**

See weaknesses

---

> ### Author Response · Authors · 2025-11-23
>
> Thank you for these comments.
>
> We agree that some of our findings resemble results of studies from the word2vec days. We believe our paper’s contributions are nevertheless important for two reasons. First, the intuition common for word2vec models – that concepts’ geometric alignment in latent space is proportional to their relatedness of meaning – is not commonplace in the age of LLMs. Instead, much of the discourse around feature geometry in LLMs emphasizes the concept of “superposition,” which implies that features are only non-orthogonality is an undesirable byproduct of compression. One goal of this paper is to bring word2vec intuitions back into consideration for LLMs.
>
> You mention that Park et al. specify that their results only apply to “causally separable concepts.” We agree, but ask which concepts are “causally separable?” A core part of our argument is that many features that are semantically distinct (like goodness and beauty) are not causally separable. So while we agree that whitening is fine if the concepts are causally separable (perhaps by definition), an aim of our paper is to show that many of the semantic features we might want to monitor or steer are not causally separable.
> Finally, we see why our argument that semantics sit on a low dimensional subspace may seem unsubstantiated given that we begin by considering only 28 features, not the full space. We certainly do not intend to argue that token embeddings can be reduced to 3 dimensions with little loss of information. We want only to make claims about the *semantic* subspace of the model. While this would be yet more persuasive if we had a bank of 1000 semantic features, survey measurement is costly and we use the best data available.
>
> We now move on to the detailed points:
>
> Our discussion of representing nearly orthogonal vectors is referencing a property of high-dimensional spaces related to the Johnson-Lindenstrauss lemma. Define two unit vectors as "almost orthogonal" if the angle between them differs from orthogonality by at cos(θ) = t (for some small t). It follows that two random unit vectors in Rn are “almost orthogonal” with probability approximately 1- e^(-nt²) for large n. For example, Gemma-12B’s token embedding has dimensionality of 3840. This means the probability that two random vectors would have a cosine similarity greater than 0.1 is 2.1e-17. A succinct proof can be found here: https://www.cs.princeton.edu/courses/archive/fall14/cos521/lecnotes/lec11.pdf
> In the camera-ready version, we would be happy to formalize this idea of “almost orthogonal” with an equation to improve clarity.
>
> You ask “why is it good/bad that word embeddings are partly aligned with several reference vectors.” We don’t claim this is inherently good or bad. Our aim is to show that (i) there is substantial alignment between features, and (ii) this alignment mirrors the correlational structure found in surveys of semantic associations. An alternative hypothesis is that models represent their features as roughly orthogonal vectors because this would minimize unwanted interference between them. We try to show that this hypothesis is wrong, and that, although it would be mathematically possible for all commonly used features to be represented orthogonally, this is not what happens in LLMs. We make no claims about “good or bad,” we are simply attempting to make a scientific contribution to the understanding of LLM feature space.
>
> We ranged the color scale 0.0-0.3  because cosine similarities of 0.2 and 0.3 are substantial and important, but would be hard to perceive if the scale ranged from 0 to 1. But given that multiple reviewers disliked Figure 2, we plan to replace these heatmaps with scatterplots representing the same values. For each scatterplot, points will represent feature pairs, on the x-axis will be the feature-pair’s correlation in the survey data and on the y-axis will be the feature-pair’s correlation in the token projections (or the cosine similarity of the feature pair). Representing these values using x and y coordinates instead of color will have multiple benefits: First, it will be easier to put the cosine similarities on a scale from 0.0 to 0.3 because features paired with themselves (cos(θ)=1) need not be plotted. Second, a scatterplot would make the correlation between survey and embedding statistics much clearer than eyeballing two heatmaps. We intend to make this change in the camera-ready version unless the reviewers object.
>
> Section 5 is necessarily short due to page constraints. Changing Figure 2 from heatmaps to scatterplots should shrink the figure considerably, freeing up space to add more clarifying exposition in the camera-ready version.
>
> Finally, you suggest that our findings may only have a niche audience. We believe our claims offer a re-conceptualization of how feature space is structured. The question of “how do LLMs represent semantics” seems very general to us, so we expect a broad audience could take interest.

---

### Official Review · Reviewer_mumt · 2025-11-01

**Soundness:** 2
**Presentation:** 3
**Contribution:** 2
**Rating:** 4
**Confidence:** 2

**Summary:**

This paper uses well-established research in psychology about human semantic scales to study the internal semantics encoded in the embedding matrices of LLMs. These semantic scales experiments consist of mapping words to key semantic axes, such as “kind-cruel”. The authors consider 28 such axes (which come from the original experiment in social psychology).

First, the authors find that the feature directions from the embedding matrices corresponding to these 28 axes correlate well with the human ratings of the words (on these various semantic scales).

Second, the authors find that these LLM embedding matrices can also be “summarized” with low-dimensional matrices and that this 3-dimensional subspace roughly corresponds to the three latent dimensions of Evaluation, Potency, and Activity, that were found for humans decades ago. They do so by applying PCA to the embedding matrix.

Third, the authors study the question of model behavior and steering through these same matrices. They test whether intervening on one feature (e.g., “soft-hard”), will also have an effect on other axis, where the intervention is carried out on the model’s token embeddings. They find that the magnitude of these off-target effects is proportional to the cosine similarity between the vectors.

**Strengths:**

- It is important to study how LLMs understand semantics and how their internal (low-dimensional) representations work. Using psychology studies and well-established theories on human semantics is a valuable way of doing so, and these kinds of interdisciplinary approaches seem particularly relevant
- The different kinds of experiments nicely build on each other and feel like a natural progression. First they study the correlation between the human ratings and token embeddings, then they perform PCA and go into low-dimensions (specifically, into a 3-dimensional solution), and lastly they try to steer the model and predict off-target effects.
- It is a good idea to test the “whitened” embeddings method in order to ensure that the experimental results are more robust
- The paper is well-written and the authors clearly have a great command of the social psychology literature at hand
- The experiments are tested across multiple different LLMs

**Weaknesses:**

I have three main criticisms of the framework and experimental set-up of the paper.
- First, the paper only focuses on the LLM static embedding matrices, without taking the activations into account. The authors justify this in page 2, saying that “embedding and unembedding matrices also warrant attention”, but it seems too simplifying of an assumption to forget about the context and the actual transformer architecture of the LLM. Particularly because all of the experiments are run on LLMs, the context matters a great deal.
- Second, and related to the first point, it seems like the methods of the paper are meant to apply to older word embedding approaches like word2vec, but aren’t really updated to today’s architectures. In order words, the experiments are run on LLMs, but the framework of the paper and the set-up of “good-bad” pairs of antonyms seems to be taken directly from the BERT era.
- Third, I think that Figure 3 isn’t fully demonstrating the conclusions that are claimed. Specifically, the Evaluation, Potency, and Activity separation isn’t that convincing, and the authors also observe this by saying that “the second component admittedly does not correspond to Potency”. Moreover, the results for all of the figures are not really generalizing across different LLMs.

**Questions:**

- See the three main points made in the “weaknesses” section
- Isn’t it clear a priori that the off-target effect is proportional to the cosine similarity? Why is this an important finding?
- Given the initial motivations of the Evaluation, Potency, and Activity studies, which were found to generalize across languages, have you thought about replicating your studies to LLMs trained with other languages other than English?
- Why has the code not been included as part of the submission?

---

> ### Author Response · Authors · 2025-11-23
>
> Thank you for these comments. We are glad you recognize the contributions of this work. We respond below to the weaknesses and questions you raised.
>
> First, you note that we only analyze the token embeddings rather than the activations in the hidden layers. While we agree that an analysis of activations on the model’s middle layers would be interesting, we believe this study provides groundwork for such future analyses. As we mention in the text, the initial embedding layer has the benefit of having its vectors directly associated with tokens, making them more readily interpretable and making features more straightforward to extract. Second, they enable analyses into the ways that features are encoded in the *weights* of the model. The semantic associations encoded in the token embedding therefore influence all generations including those tokens. While we agree that the activation patterns in the models’ internals are also interesting and important (and a direction we are pursuing in current work!), we believe the weights in token embeddings are also important grounds for analysis.
>
> On a related point, mention that our methods more closely resemble work from the word2vec days than more recent work with LLMs. This is true, and is a result of analyzing the token embeddings instead of activations. Activations are extracted and intervened upon differently than token vectors. Practically speaking it would be difficult to include an analysis of both weights and activation patterns in a single paper, as the features are extracted and intervened upon differently. We therefore believe this paper offers initial steps that motivate follow-up work that explores the geometry of semantic features in activation space.
>
> You ask about the EPA dimensions, and question the correspondence between the embedding’s loadings and those from the survey. As you note, we state in the text that the second principal component differs somewhat, resembling “beautiful vs. plain” more than “strong vs. weak,” but the first and third components consistently capture “good vs. bad” and “active vs. passive” quite accurately. We believe that, while this is not identical to human survey ratings structure, the similarity is still notable. You also claim that “the results for all of the figures are not really generalizing across different LLMs.” It is true that the raw correlations within the heatmaps are weaker for the Llama and Qwen models, but the most important findings: (1) association between survey correlations and feature projection correlations, (2) PCA structure, and (3) correlation of off-target effects with cosine similarity, are similar between models. The findings are clearest in the Gemma models, and that is why we foreground them, but the same basic patterns hold across model families. If there is a specific finding that you believe does not generalize across models, could you please point it out?
>
> Lastly, you ask “Isn’t it clear *a priori* that the off-target effect is proportional to the cosine similarity? Why is this an important finding?” We agree that these results are not totally surprising given the earlier results, but the analyses are intended to show that position on feature vectors is not merely correlated with semantic association but causally affects the word’s semantics during inference. We agree that it is intuitive that the relation would be causal and not just correlational, we believe it is still worth showing this empirically. And given that these vectors pass through dozens of non-linear transformations and interactions during the forward pass, we do not think the causal relation can be derived a priori from the correlation.
>
> Finally, you ask about the availability of code. Our codebase was not in shareable shape at the time of submission, but we will happily share a link to shareable code in the camera-ready version.
>
> Thank you again for these thoughtful comments.

---

### Meta-Review · Area_Chair_BJyh · 2025-12-29

**Summary:**

This paper uses methods from psychology to evaluate the semantic structure of of word embeddings in LLMs, i.e., this work focuses on evaluating existing models, not proposing new methodologies or mathematical analyses.

The methods of the paper are meant to apply to older word embedding approaches like word2vec, not to today’s architectures.

It is unclear how these insights could be applied to improve LLM architectures or their downstream performance.

**Reviewer Concerns:**

Minor formatting mistakes have all been corrected.

**Reviewer Scores:**

None is likely to change.

---

### Decision · Program_Chairs · 2026-01-26

Reject